# Stability of Class II Malocclusion Treatment with the Austro Repositioner Followed by Fixed Appliances in Brachyfacial Patients

**DOI:** 10.3390/ijerph18189793

**Published:** 2021-09-17

**Authors:** Maria Dolores Austro-Martinez, Ana I. Nicolas-Silvente, Eugenio Velasco-Ortega, Alvaro Jimenez-Guerra, Jose A. Alarcon

**Affiliations:** 1Department of Restorative Dentistry, School of Dentistry, CEIR Campus Mare Nostrum, University of Murcia, 30008 Murcia, Spain; dra.austro@gmail.com; 2Dental Pathology and Therapeutics, School of Dentistry, CEIR Campus Mare Nostrum, University of Murcia, 30008 Murcia, Spain; 3Department of Comprehensive Dentistry for Adults and Gerodontology, University of Seville, 41009 Seville, Spain; evelasco@us.es (E.V.-O.); alopajanosas@hotmail.com (A.J.-G.); 4Section of Orthodontics, Faculty of Odontology, University of Granada, 18071 Granada, Spain; jalarcon@ugr.es

**Keywords:** angle class II, brachyfacial patients, fixed functional appliance, mandibular retrusion

## Abstract

One of the goals of functional-appliance devices is to modify the vertical growth pattern, solving several kinds of malocclusion. This study aimed to evaluate Class II malocclusion treatment’s stability with Austro Repositioner, followed by fixed appliances, and assess its capacity to modify vertical dimensions in brachyfacial patients. A test group of 30 patients (16 boys and 14 girls, mean 11.9 years old) with Class II malocclusion due to mandibular retrognathism and brachyfacial pattern treated with Austro Repositioner and fixed appliance were compared to a matched untreated Class II control group of 30 patients (17 boys and 13 girls, mean age 11.7 years old). Lateral cephalograms were taken at T1 (initial records), T2 (end of treatment), and T3 (one year after treatment). Statistical comparisons were performed with a paired-sample *t*-test and two-sample *t*-tests. Significant improvements in the skeletal Class II relationship were observed in the treated group. The ANB angle decreased (4.75°), the SNB angle increased (3.92°), and the total mandibular length (Co-Pg) increased (8.18 mm) (*p <* 0.001). Vertical dimensions were also significantly modified, the FMA angle increased (3.94°), LAFH-distance increased (3.15 mm), and overbite decreased (3.35 mm). These changes remained stable one year after treatment. The Austro Repositioner was adequate for treating the skeletal Class II malocclusion resulting from the mandible retrusion in brachyfacial patients.

## 1. Introduction

Class II angle malocclusion is a widespread skeletal discrepancy. In most cases, it is a significant component of mandibular retrognathism [1,2], so functional appliances are highly recommended for treatment. Several studies have already evaluated skeletal, dentoalveolar, and facial changes after treatment with functional devices in Class II, division I patients with mandibular retrognathism [3,4,5,6]. However, the evaluation of the long-term stability in these changes is scarcer [7,8,9].

One of the most interesting aspects to consider regarding functional appliances is their ability to modify the vertical growth pattern, with it being an essential tool for the therapeutic approach of this malocclusion. A previous study about a recent fixed functional appliance, the Austro Repositioner, found encouraging short-term results when applied to brachyfacial skeletal Class II patients [10]. The results showed significant improvement in skeletal Class II growth, resulting from a significant mandibular plane angle increase and overbite decrease. Functional devices try to control facial vertical growth by acting on the vertical position of the molars. In the case of the Austro Repositioner, the mechanism that induces favorable changes in the facial pattern of brachyfacial patients is due to the extrusion of posterior-lower molars that it induces; this causes a mandibular postero-rotation and a reduction in the overbite. In addition, no patient collaboration is required as it is a fixed appliance that can be combined with other fixed devices simultaneously.

To date, only a few studies have described vertical changes in brachyfacial Class II patients treated with other functional appliances. Studies about treatment with Forsus [11] and Herbst [12,13] appliances reported only a slight effect on the mandibular plane angle. On the contrary, vertical dimension changes can be achieved after treatment with Twin-Block [14].

The ability of some fixed devices such as Forsus or Herbst to increase the mandibular plane angle in patients with a marked brachyfacial pattern is, therefore, limited. Conversely, with the Twin-Block, the mandibular plane angle can be improved, but the patient’s collaboration is required as it is a removable appliance. In addition, if a subsequent phase with a fixed device is needed, the total duration of treatment is extended.

This study aimed to evaluate the skeletal and dentoalveolar stability of Class II malocclusion treatment with the Austro Repositioner, followed by fixed appliances, and assess its capacity to modify vertical dimension brachyfacial patients.

## 2. Materials and Methods

### 2.1. Samples

This prospective clinical study was based on the records of 60 patients with skeletal Class II malocclusion due to mandibular retrognathism and brachyfacial pattern. The present study received a favorable report from the Research Ethics Commission of the University of Murcia in June 2015. All patients were evaluated in 3 different stages: before treatment (T1), after two phases of functional orthopedic and fixed appliances treatment (T2), and one-year follow-up after treatment (T3), and distributed into two groups:-Experimental group: 30 consecutively treated patients, selected from a private practice and treated by the same orthodontist (M.D.A.-M.). The sample was composed of 16 boys and 14 girls, with a mean age of 11.9 (at T1), 14.3 (at T2), and 15.7 (at T3).-Control group: 30 untreated patients selected from the online Craniofacial Growth Legacy Collection (http://www.aaoflegacycollection.org, accessed on 20 February 2015), which consists of several well-known growth studies such as the Burlington and Michigan growth studies. The sample was composed of 17 boys and 13 girls, with a mean age of 11.7 (at T1), 14.3 (at T2), and 15.6 (at T3).

Informed consent was signed from the parents of all patients included in the study.

Lateral cephalograms were acquired in each stage.

#### 2.1.1. Inclusion Criteria

All subjects selected presented skeletal Class II malocclusion (ANB angle > 5°) resulting from the retrusion of the mandible (SNB angle < 78°), brachyfacial pattern (FMA angle < 20°), overjet ≥ 5 mm, symmetric Class II molar relationship (minimum severity of one-fourth Class II molar relationship; mean amount of Class II molar relationship was 6.5 ± 2.23 mm), and skeletal growth maturation between stages 3 and 4, according to the cervical vertebral maturation (CVM) method [15], at the beginning of treatment. In short, the morphology of the vertebrae was assessed, accounting for the fact that in CS3, the lower edges of C2 and C3 show a concavity, and that at least one cervical body had a trapezoidal shape. In CS4 it was observed that all the lower edges of C2, C3, and C4 show concavity and that both C3 and C4 have a rectangular shape. One of the inclusion criteria of our study was that all the cases of both groups were between the CS3 and CS4 stage, which is when the peak of growth takes place.

#### 2.1.2. Exclusion Criteria

Patients with congenitally missing or extracted permanent teeth (except third molars), posterior crossbites or severe maxillary transverse deficiency, severe facial asymmetry determined by clinical or radiographical examination, congenital syndromes, TMJ disorder, previous orthopedic/orthodontic treatment, and poor oral hygiene were excluded from the study.

### 2.2. Class II Malocclusion Treatment Protocol

Subjects were treated in two phases. The first phase comprised the Austro Repositioner appliance, according to a previously described protocol [10] (Figure 1), specially designed by brachyfacial patients; in brief, the acrylic resin wedge was extended to the upper incisors, and in this anterior segment, the wedge was 1.0–1.5 mm thicker.

Due to this specific design, the lower incisors make contact with the anterior area of the acrylic resin wedge and prevent the occlusion of the posterior teeth, thus, promoting their eruption and improving the overbite (Figure 2), and can be used in combination with fixed appliances (Figure 3).

After a minimum of a 1 year mean period of functional treatment, orthodontic treatment continued in a simultaneous second phase using fixed appliances (0.022-inch slot conventionally ligated Hilgers’ edgewise bracket system; Ormco, Glendora, Calif), with both Austro Repositioner and fixed appliances for 2 to 2.5 years, followed by a 1 year of retention period with a removable standard Hawley-type retainer (6 months full-time, 6 months nighttime) in the upper arch, and a canine-to-canine fixed lingual retainer in the lower arch. No class II elastics were worn during the multibracket phase.

### 2.3. Measurement Method

For all subjects, standard lateral cephalometric radiographs with the teeth in centric occlusion and the head oriented horizontally with the Frankfort plane were acquired according to a previously described protocol [10]. Linear and angular measurements from the analyses of Steiner [16], Ricketts [17], and McNamara [18] (SNA (°), SNB (°), ANB (°), Pt A-Na perp (mm), Pg-Na perp (mm), Co-Pg (mm), FMA (°), LAFH (mm), overbite (mm), overjet (mm), U1 to SN (°), L1 to GoMe (°), interincisive angle (°), were measured using Dolphin Imaging 11.0 software (Chatsworth, CA, USA).

### 2.4. Statistical Analysis

The sample size was established with two simple *t*-tests, a study power of 90%, a significant level of 0.05, and a detected difference of 1.2°, based on a previous study [10]. The results showed that a minimum of 20 patients was needed in each group.

A standard statistical software package (IBM SPSS statistics 20, IBM Armonk, New York, NY, USA) was used for statistical analysis. After confirmation of a normal distribution of the variables, the differences in the pretreatment variables between the two groups were determined using a two-sample *t*-test. Intra-groups changes were evaluated by using a paired-sample *t*-test comparing the pre-and post-treatment and post-treatment-1-year values. Inter-groups comparison changes were determined using a two-sample *t*-test.

All images were scored by a single, experienced observer (M.D.A.-M.). To test for observer reliability, 30 randomly selected images were reassessed by the same observer and scored by another independent expert (J.A.A.). Inter-and inter-rater agreements were calculated using Cohen’s kappa (ƙ) coefficient [19].

## 3. Results

The inter- and intra-rater agreement coefficients were κ = 0.91 and κ = 0.90, respectively.

There were no statistically significant differences between the treated and control groups at T1 (Table 1).

Table 2 shows an inter- and intra-groups comparison of the treatment or observation period changes (T2–T1). In the treated group, a significant decrease in the ANB angle was found. No significant differences were found in the maxillary skeletal measurements. In contrast, the SNB angle showed an increase of 3.92° in the treated group, compared with 0.85° in the controls (*p* < 0.001).

The improvement in the skeletal Class II observed in the treated group can be attributed to changes in the mandible, as no significant differences were observed in the maxillary cephalometric measurements (SNA angle and Pt A-Na perp). In contrast, the SNB angle, Pg-Na perp, and Co-Pg distance increased significantly in the treated group compared to the control group (*p <* 0.001). Mandibular skeletal changes also caused a significant reduction in the overjet in the treated group.

Vertical changes showed a significant increment in the FMA angle, and in the LAFH distance, and a reduction in the OB in the treated group, reflecting a modification in the vertical dimensions after treatment with Austro Repositioner in brachyfacial Class II patients. Finally, retroclination of both the upper and lower incisors was found after treatment.

Table 3 shows intra- and inter-group differences during the post-treatment and one-year post-treatment/observation period (T3–T2); no significant changes were observed in the treated group, while differences between groups persisted during this period, showing the stability of treatment results.

The facial and intraoral features at T1 are shown in Figure 4 and at T2 in Figure 5.

## 4. Discussion

The outcomes and stability of results with the Austro Repositioner and fixed appliances in patients with Class II malocclusion due to mandibular retrognathism and brachyfacial growth pattern have been analyzed in the present study, paying particular attention to the vertical dimension.

All patients wore the Austro Repositioning fixed functional appliance throughout treatment, wearing only the Austro Repositioning appliance during the first year of treatment, and then this functional appliance together with fixed multi-braces appliances until the end of treatment. The subjects of both groups were between the CS3 and CS4 stage of skeletal maturation, which is when the peak of growth takes place. The literature concerning treatment timing for Class II malocclusion with functional appliances has revealed that the treatment is most effective when it includes the peak in mandibular growth [15].

Patients were evaluated after active treatment and one year later. After the active treatment phase (T2–T1), a significant improvement in skeletal Class II was observed due to a substantial increase in mandibular growth, a reduction in overjet, and a retroinclina-tion of the upper and lower incisors. In addition, there were significant changes in the vertical dimension, with an increase in the FMA angle and LAFH distance and a reduction in the overbite. All these changes remained stable during the post-treatment evaluation (T3–T2) so that, one year later, we found stability in the results achieved by the treatment.

The results observed in the present study regarding the stability at one-year follow-up confirm the effects observed in the short term [10] and, therefore, indicate that the Austro Repositioner is an effective fixed functional device for treating skeletal Class II malocclusion with a mandibular origin. It also allows the vertical dimension to be modified, so it would be especially indicated for treatment of patients with a brachyfacial growth pattern.

The literature is scarce in specific studies on the stability of treatment results with fixed functional appliances applied to brachyfacial patients and their ability to modify vertical dimensions.

One of the most significant clinical interest findings in our study is the high increase in the SNB angle observed after treatment, which remains stable in the one-year follow-up. An increased SNB angle has been found in other studies with various functional appliances [20,21]; nevertheless, comparisons with the results obtained in our study must be made with caution, because these studies included patients with different facial growth patterns (i.e., meso and brachyfacial patterns). In the evaluation developed by Gunay et al. [22] on mesofacial and brachyfacial patients, no significant changes were found in SNB angle for late adolescent patients treated with Forsus. In our study, the ANB angle was considerably reduced in the treated group (4.75° on average). We did not find significant changes during the one-year follow-up post-treatment, reflecting stability in the improvement in skeletal Class II achieved with this treatment. The reduction in the ANB angle observed in our study is much higher than that reported in other studies with different functional devices [8,20].

As reported before, the improvement in the skeletal Class II in patients treated with Austro Repositioner can be exclusively attributed to changes in the mandible. Both the position of the Pg and the total mandibular length (Co-Pg) experienced a significant increase in our patients, which was considerably higher than that reported by other studies with fixed [4,20,23,24] and removable appliances [14]. However, other authors found no significant skeletal effects on the jaw after treatment with Herbst appliance [12] or Forsus appliance [11,22].

One of the main interests in the design of our study was to evaluate the ability of the Austro Repositioner to modify the vertical dimensions in brachyfacial patients in a stable way. The results indicate that the favorable changes in the vertical parameters (significant increase in FMA and LAFH and reduction in OB) remain stable in the one-year follow-up evaluation. This therapeutic possibility is much more limited with other fixed functional appliances, where there are no significant changes in the vertical pattern [11,12,13,20,20,25]. The vertical dimensions can be manipulated with the Twin Block, at least in the short term [14], although, as it is a removable appliance, patient compliance is mandatory. In contrast, the Austro Repositioner does not require patient compliance and can be worn simultaneously with fixed braces appliances to reduce the final treatment time.

Regarding dental changes, the most common difference found in the literature is the retroinclination of the upper incisors and the proinclination of the lower incisors, and consequently, a reduction in the overhang after treating with functional appliances, both removable and fixed [6,14,20,22,26,27,28]. In our study, if there was a retroinclination of the upper incisor, the lower incisors were also retroinclined. No changes were seen in the position of the lower incisors between T2 and T3, which indicates stability in the overjet correction due to skeletal effect on the jaw. The mandibular growth observed during treatment remained stable one year later; there were no changes in the position of the lower incisors that could have masked this skeletal effect. These data reflect that the skeletal changes observed in the mandible played an essential role in reducing the overjet observed in the patients treated with the Austro appliance. The greater correction achieved in skeletal Class II, as well as its ability to modify vertical dimensions at least one year after treatment and retention period, compared to other functional devices, led us to consider the Austro Repositioner as an effective option for the treatment of skeletal Class II patients with a mandibular origin and brachyfacial growth pattern.

The limitations of the present study include the short post-treatment evaluation period (only 1 year), and the sample size (larger samples would be desirable), that limits a deeper study of the differences in treatment with Austro Repositioner according to sex.

It would be desirable to carry out more studies evaluating longer follow-up periods and the effect of the Austro Repositioner appliance in patients with different growth patterns.

## 5. Conclusions

Within the limits of the present short-term follow-up clinical study (1 year post-treatment), we can conclude that the use of the Austro Repositioner appliance was adequate for treating skeletal Class II malocclusion resulting from the retrusion of the mandible in brachyfacial patients.

## Figures and Tables

**Figure 1 ijerph-18-09793-f001:**
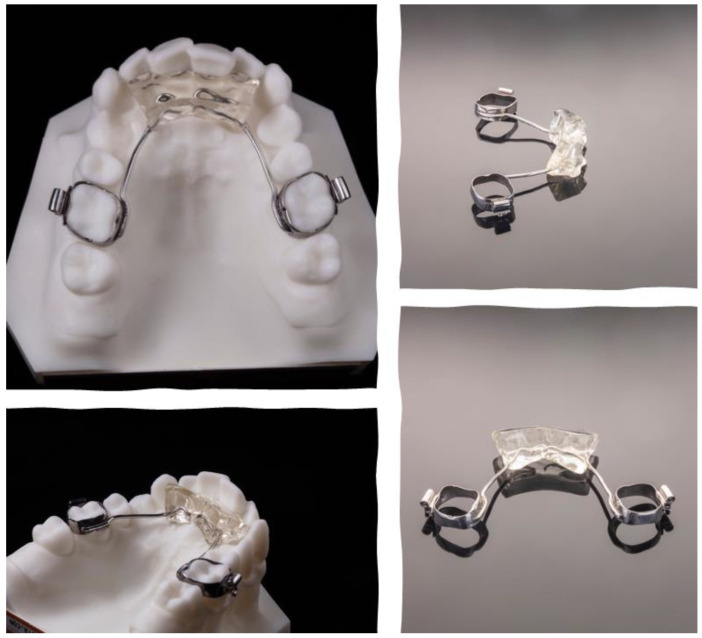
Austro Repositioner appliance design.

**Figure 2 ijerph-18-09793-f002:**
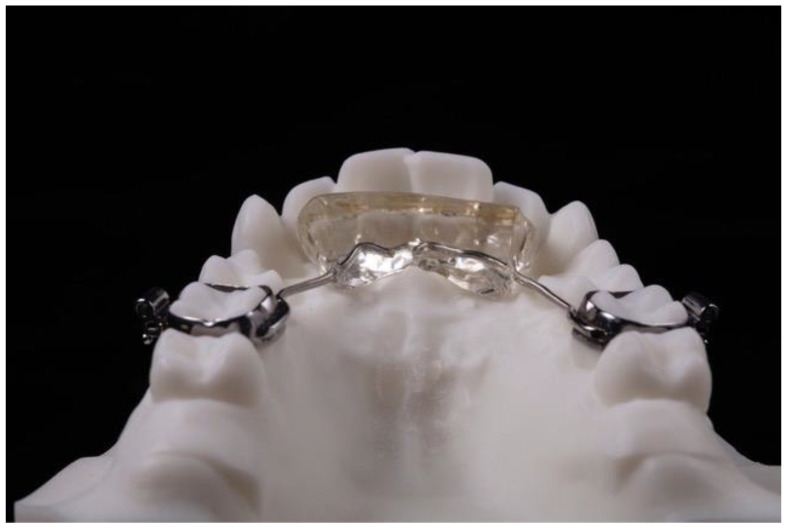
Acrylic resin wedge in the lower incisors.

**Figure 3 ijerph-18-09793-f003:**
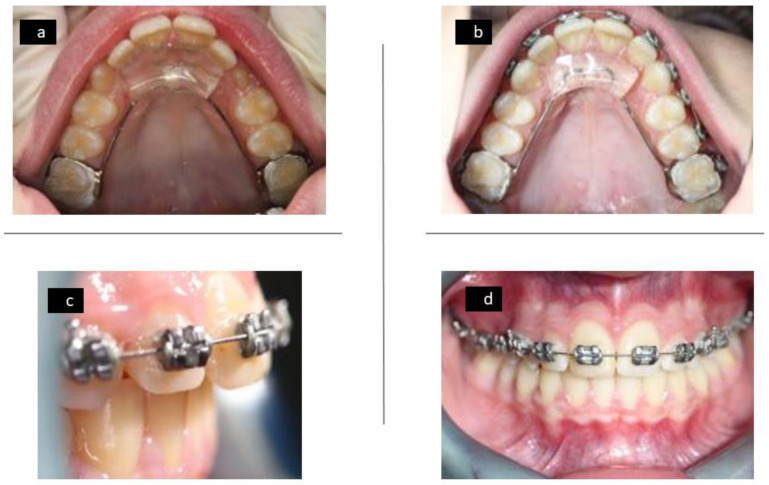
Upper occlusal photograph with fixed Austro Repositioner appliance (**a**), upper occlusal photograph with fixed Austro Repositioner appliance combined with braces (**b**), lateral anterior photo with fixed Austro Repositioner appliance combined with braces (**c**), and frontal photo with fixed Austro Repositioner appliance combined with braces (**d**).

**Figure 4 ijerph-18-09793-f004:**
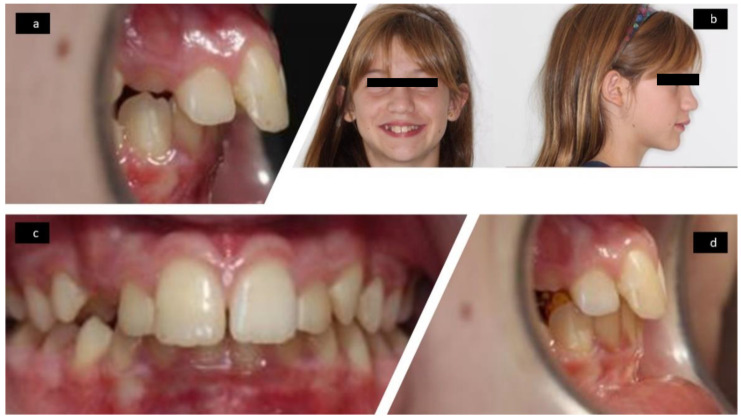
Photographs at T1: lateral intraoral without Austro Repositioner appliance (**a**), frontal and lateral facial (**b**), frontal intraoral (**c**), lateral intraoral with Austro Repositioner appliance (**d**).

**Figure 5 ijerph-18-09793-f005:**
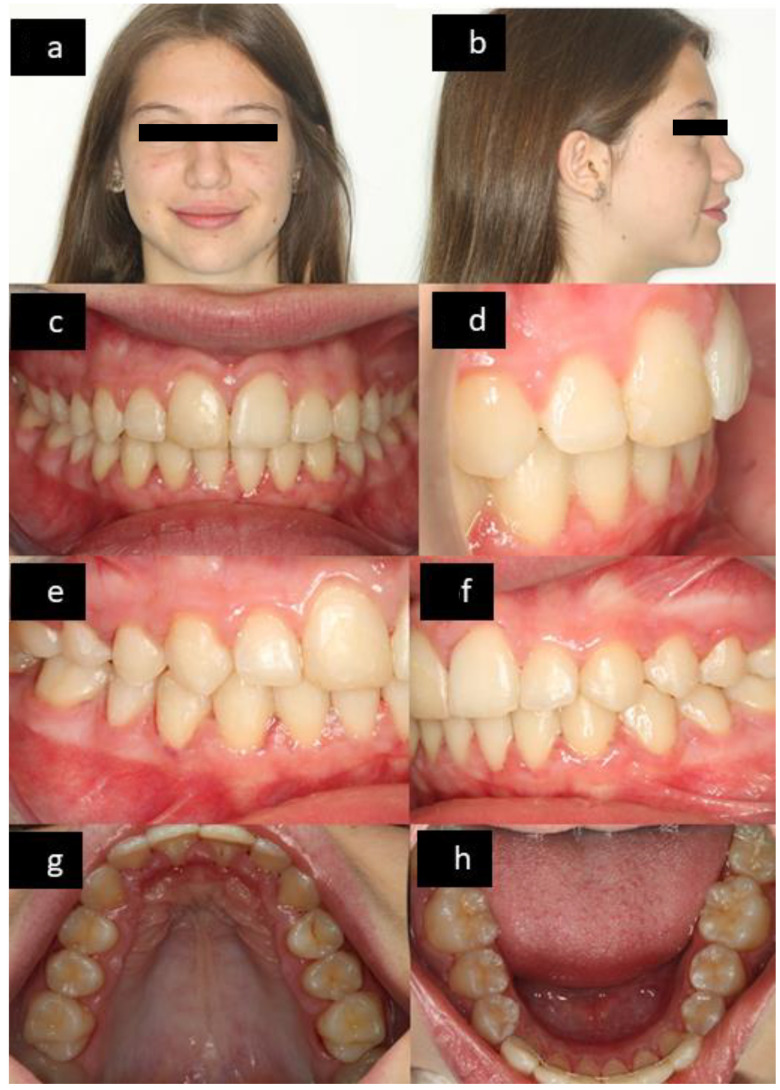
Photographs at T2: frontal facial (**a**), lateral facial (**b**), frontal intraoral (**c**), lateral anterior intraoral (**d**), right lateral intraoral (**e**), left lateral intraoral (**f**), upper occlusal (**g**), and lower occlusal (**h**).

**Table 1 ijerph-18-09793-t001:** Intergroup comparison at the beginning (T1).

Measurements	Treated Group		Control Gruop		*p* Value	
	Mean	SD	Mean	SD		
SNA (°)	82.1	1.84	82.07	1.77	0.827	NS
SNB (°)	75.32	1.72	75.38	1.52	0.91	NS
ANB (°)	6.72	1.31	6.68	0.98	0.545	NS
Pt A-Na perp (mm)	2.76	1.65	2.8	2.07	0.487	NS
Pg-Na perp (mm)	−2.59	1.18	−2.44	1.42	0.657	NS
Co-Pg (mm)	100.54	2.24	100.51	2.17	0.994	NS
FMA (°)	19.41	0.27	19.47	0.31	0.495	NS
LAFH (mm)	53.93	2.47	53.98	1.13	0.838	NS
OB (mm)	6.01	1.57	6.04	2.45	0.568	NS
OJ (mm)	6.81	1.12	6.86	1.43	0.375	NS
U1 to SN (°)	103.56	2.27	103.08	2.81	0.627	NS
L1 to GoMe (°)	93.11	1.41	93.23	1.43	0.281	NS
Interincisal Angle (°)	127.01	3.51	127.97	3.37	0.933	NS

**Table 2 ijerph-18-09793-t002:** Inter and intra-groups comparison of the treatment/observation period changes (T2–T1).

Measurements	Treated Group				Control Group				Treated vs. Control	
	Mean Differences	SD	*p* Value		Mean Differences	SD	*p* Value		*p* Value	
SNA (°)	−0.19	1.31	0.983	NS	0.06	1.77	0.639	NS	0.673	NS
SNB (°)	3.92	1.28	<0.001	S	0.85	1.52	<0.001	S	<0.001	S
ANB (°)	−4.75	0.85	<0.001	S	−0.64	0.98	0.001	S	<0.001	S
Pt A-Na perp (mm)	0.22	1.29	0.371	NS	−0.34	2.07	0.231	NS	0.117	NS
Pg-Na perp (mm)	−5.17	1.74	<0.001	S	−1.64	2.42	<0.001	S	<0.001	S
Co-Pg (mm)	8.18	3.23	<0.001	S	2.36	2.17	<0.001	S	<0.001	S
FMA (°)	3.94	2.04	<0.001	S	−0.22	1.38	0.031	S	<0.001	S
LAFH (mm)	3.15	1.31	<0.001	S	0.5	1.13	0.029	S	<0.001	S
OB (mm)	−3.35	1.59	<0.001	S	0.16	1.45	0.031	S	<0.001	S
OJ (mm)	−4	2.11	<0.001	S	−0.64	2.43	<0.001	S	<0.001	S
U1 to SN (°)	−2.06	1.82	<0.001	S	−0.56	2.81	0.048	NS	<0.001	S
L1 to GoMe (°)	−2.59	2.26	<0.001	S	0.36	1.34	0.522	S	<0.001	S
Interincisal Angle (°)	4.17	3.91	<0.001	S	−0.53	3.37	0.002	S	<0.001	S

**Table 3 ijerph-18-09793-t003:** Intra- and inter-groups differences during the posttreatment and one-year post-treatment/observation period (T3–T2).

Measurements	Treated Group				Control Group				Treated vs. Control	
	Mean Differences		*p* Value		Mean Differences	SD	*p* Value		*p* Value	
SNA (°)	0.5	1.79	0.877	NS	0.21	1.56	0.251	NS	0.531	NS
SNB (°)	1.2	1.02	0.407	NS	0.92	1.63	0.047	NS	0.001	S
ANB (°)	−0.9	0.52	0.818	NS	−0.5	0.87	0.039	NS	<0.001	S
Pt A-Na perp (mm)	0.33	1.64	0.652	NS	0.5	1.86	0.357	NS	0.322	NS
Pg-Na perp (mm)	0.74	1.48	0.431	NS	0.93	2.8	0.045	NS	0.031	S
Co-Pg (mm)	0.88	3.39	0.24	NS	0.71	2.1	0.037	NS	<0.001	S
FMA (°)	0.14	1.86	0.442	NS	0.35	1.38	0.033	NS	0.023	S
LAFH (mm)	0.12	2.45	0.145	NS	0.82	1.16	0.349	NS	<0.001	S
OB (mm)	0.14	1.38	0.451	NS	0.13	1.42	0.041	NS	<0.001	S
OJ (mm)	1.00	1.54	0.121	NS	0.87	0.5	0.032	NS	<0.001	S
U1 to SN (°)	0.61	2.52	0.371	NS	0.12	2.62	0.049	NS	0.033	S
L1 to GoMe (°)	0.53	2.56	0.432	NS	0.33	1.45	0.22	NS	0.021	S
Interincisal Angle (°)	0.13	3.37	0.139	NS	0.45	3.28	0.225	NS	0.014	S

## Data Availability

The data presented in this study are available on request from the corresponding author. The data are not publicly available due to privacy.

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
