# Peer review of "Stability of Class II Malocclusion Treatment with the Austro Repositioner Followed by Fixed Appliances in Brachyfacial Patients"

_ijerph, 2021, doi:10.3390/ijerph18189793_

Round 1
Reviewer 1 Report
A short description of the CVM measurements would be appropriate. All treated and control cases seem to be mandibular class II, lower incisor position is the key of the functional treatment. This is why I would consider important the presentation of dental changes as well in T2 and T3 phases.
Author Response
REPLY TO REVIEWER # 1
Thank you very much for your comments and the review you made of our research.
- Concern of the reviewer:
“A short description of the CVM measurements would be appropriate”
Our response:
A brief description of the followed CVM method has been included in the material and method section:
“In short, the morphology of the vertebrae has been assessed taking into account that in CS3, the lower edges of C2 and C3 show a concavity, and that at least one cervical body has a trapezoidal shape. In CS4 it is observed that all the lower edges of C2, C3 and C4 show concavity and that both C3 and C4 have a rectangular shape. An inclusion criteria of our study was that all the cases of both groups were between the CS3 and CS4 stage, which is when the peak of growth takes place.”
- Concern of the reviewer:
“All treated and control cases seem to be mandibular class II, lower incisor position is the key of the functional treatment. This is why I would consider important the presentation of dental changes as well in T2 and T3 phases”
Our response:
We agree with the reviewer, lower incisor position is the key of the functional treatment. The presentation of dental changes in T2 and T3 phases are showed in table 3 (“Table 3. Intra and inter groups differences during the posttreatment and one-year posttreatment / observation period (T3-T2).”
A new paragraph with the interpretation of these results has been included in the discussion section:
“No changes were seen in the position of the lower incisors between T2 and T3, which indicates stability in the overjet correction due to skeletal effect on the jaw. The mandibular growth observed during treatment remained stable one year later; there were no changes in the position of the lower incisors that could have masked this skeletal effect.”

Reviewer 2 Report
The article IJERPH-135682 entitled” Stability of Class II malocclusion treatment with the Austro Repositioner followed by fixed appliances in brachyfacial patients” showed an attractive appliance with excellent results compared to a matched untreated Class II control group patients. For this manuscript with better quality, there are still some points needed to be clarified.
Specific
- Would you please use the newest IJERPH Microsoft Word template?
- This manuscript has a considerable percentage of repeatable sentences/paragraphs with the other article (Ref. 10. Penalver, M.A.; Perez, D.; Alarcon, J.A. Short-term dentoskeletal changes following Class II treatment using a fixed functional appliance: the Austro Repositioner : A pilot study. J Orofac Orthop 2018, 79, 147-156, doi:10.1007/s00056-018-0135-3).
- Introduction, please add a paragraph to describe how the general functional appliance induces the vertical changes in introduction; and clarify the advantage and specific points from Austro Repositioner.
- Material and methods, please add the treatment and train protocol of the Autro Repositioer.
- Material and methods: Inclusion and exclusion criteria.
- T1~T2, please discuss the effect of the functional appliance and fixed appliance, especially the sequence and period of Austra repositioner and fixed appliance.
- Material and methods, please add the result and evidence of cephalometry.
- This article includes a test group of 30 patients (16-boys and 14-girls, mean
11,9 years old), the growth and development potential of boys and girls was different. Please provide the evidence of biological age (example: maturity of skeletal from X-ray etc.) to explain the effect of Austro Repositioner.
- If the sex of the growth and development plays an essential part in this appliance, the sample size and power should be reevaluated in this manuscript.
- Discussion: Please add the comparison of the Austra repositioner with the other functional appliance
- Discussion: Please add the factor from sex, the limitation and future .
- Page 5,Result, Please correct “The inter- and interrater agreement”
Author Response
REPLY TO REVIEWER # 2
Thank you very much for your comments and the review you made of our research.
- Concern of the reviewer:
“Would you please use the newest IJERPH Microsoft Word template?”
Our response:
The newest IJERPH Microsoft Word template has been used in the revision version of the manuscript (R1)
- Concern of the reviewer:
“This manuscript has a considerable percentage of repeatable sentences/paragraphs with the other article (Ref. 10. Penalver, M.A.; Perez, D.; Alarcon, J.A. Short-term dentoskeletal changes following Class II treatment using a fixed functional appliance: the Austro Repositioner : A pilot study. J Orofac Orthop 2018, 79, 147-156, doi:10.1007/s00056-018-0135-3).”
Our response:
We agree with the reviewer; that is because the present study derives from the same research project about Austro Repositioner fixed functional appliance, and therefore the protocol used is the same. Nevertheless the percentage of repeatable sentences/paragraphs with the other article has been reduced.
- Concern of the reviewer:
“Introduction, please add a paragraph to describe how the general functional appliance induces the vertical changes in introduction; and clarify the advantage and specific points from Austro Repositioner.”
Our response:
A new paragraph has been added in the introduction section: “Functional devices try to control facial vertical growth by acting on the vertical position of the molars mainly. In the case of the Austro Repositioner, the mechanism that induces favorable changes in the facial pattern of brachyfacial patients is due to the extrusion of posterior-lower molars that it induces; this causes a mandibular postero-rotation and a reduction of the overbite.”
- Concern of the reviewer:
“Material and methods, please add the treatment and train protocol of the Austro Repositioner.”
Our response: This information is already included, please see section 2.4. Class II malocclusion treatment protocol
- Concern of the reviewer:
“Material and methods: Inclusion and exclusion criteria.”
Our response: Inclusion and exclusion criteria are already include, please see sections 2.2. Inclusion Criteria and 2.3. Exclusion Criteria
- Concern of the reviewer:
“T1~T2, please discuss the effect of the functional appliance and fixed appliance, especially the sequence and period of Austra repositioner and fixed appliance.”
Our response:
Description of the sequence and period of Austro Repositioner and fixed appliance has been clarified in the material and method section: “After a minimum of 1 year mean period of functional treatment, orthodontic treatment continued in a simultaneous second phase by using fixed appliances (0.022-inch slot conventionally ligated Hilgers' edgewise bracket system; Ormco, Glendora, Calif), with both Austro Repositioner and fixed appliances for 2 to 2.5 years, followed by a one year of retention period with a removable standard Hawley-type retainer (6 months full time, six months night time) in the upper arch, and a canine to canine fixed lingual retainer in the lower arch.”
The effect of the functional appliance and fixed appliance are discussed in the discussion section.
- Concern of the reviewer:
“Material and methods, please add the result and evidence of cephalometry”
Our response:
Section 2.3. Measurement method has been completed by adding the cephalometric variables used in the present study (SNA (°), SNB (°)ANB (°), Pt A-Na perp (mm), Pg-Na perp (mm), Co-Pg (mm), FMA (°), LAFH (mm), Overbite (mm), Overjet (mm), U1 to SN (°), L1 to GoMe (°), Angulo interincisivo (°).
Results are commented in the discussion section
8 and 9. Concern of the reviewer:
“This article includes a test group of 30 patients (16-boys and 14-girls, mean 11,9 years old), the growth and development potential of boys and girls was different. Please provide the evidence of biological age (example: maturity of skeletal from X-ray etc.) to explain the effect of Austro Repositioner.”
If the sex of the growth and development plays an essential part in this appliance, the sample size and power should be reevaluated in this manuscript.
Our response
Biological age was used, through CVM methods, as a reference for the determination of skeletal maturation. All subjects, both boys and girls, were between the stages CS3 and CS4, regardless of chronological age, so there were no differences in the state of skeletal maturation between sexes.
- Concern of the reviewer:
“Discussion: Please add the comparison of the Austro Repositioner with the other functional appliance”
Our response:
Unfortunately, there are few studies that specifically evaluate the effect of other functional devices on patients with brachyfacial growth pattern. All of them are discussed in the discussion section
- Concern of the reviewer:
“Discussion: Please add the factor from sex, the limitation and future”
Our response:
A new paragraph addressing these issues has been added at the end of the discussion section: “Limitations of the present study include the short post-treatment evaluation period (only 1 year), and the sample size (larger samples would be desirable), that limits a deeper study of the differences in treatment with Austro Repositioner according to sex.
- Concern of the reviewer:
“Page 5, Result, Please correct “The inter- and interrater agreement”
Our response:
Done: “The inter- and intrarater agreement”

Reviewer 3 Report
Dear Editors, dear Authors,
thank you for the opportunity to review the manuscript entitled "Stability of Class II malocclusion treatment with the Austro repositioner followed by fixed appliances in brachyfacial patients". Overall, the study is well conducted and written. Nevertheless, I would suggest some adjustments before publication.
The title and even Material and Method´s section are suggestive of two separate treatment phases: first the Austro appliance, followed by Multibracket appliance. However, in the discussion section (line 208) it is stated that all patients of the treated group wore the the Austro appliance the whole treatment time. I guess this information should (1) be placed in the Material and Method´s section and (2) lead to a modification of the manuscript title.
Abstract:
- line 21: the minus signs between the number of boys and girls should be removed.
- line 30: I guess a "d" is missing at the end of "LAFH distance increase".
Introduction:
- line 41: I guess again it should be "considered" instead of "considerer".
- line 52: There is a much larger amount of studies dealing with the effect of Herbst appliance treatment on the mandibular plane angle than the two studies mentioned.
Material and Methods:
- I appreciate that the severity of Class II molar relationship is given in contrast to many studies which are keeping this major information secret. Nevertheless, the inclusion criteria regarding occlusal amount of Class II is set very low (one-fourth cuspid width). Had all treated patients a symmetric amount of Class II molar relationship pretreatment? What was the mean (SD) amount of Class II molar relationship in the present sample? Please add this information.
- As mentioned above, please add the information of treatment length (mean, SD) and the fact that the Austro and Multibracket appliance were worn parallel.
- Please add the information, if class II elastics were worn during the multibracket phase.
Results:
- all tables: please make sure that all parameters are named in English language (interincisal angle).
- 4: if T1 records are the pretreatment records, why did you show a picture with the Austro appliance in situ? And why was it removed for the fourth picture after eruption of the lower right canine (large overjet, canines about ½ cusp widths class II relationship)
Discussion:
- line 208: see above
- lines 223-224: as mentioned above, there are studies on the stability of treatment results of fixed functional appliances dealing with vertical dimensions.
- 225-229: as the present study found/assessed no specific vertical changes of the maxilla, the discussion of the two study results is unnecessary.
- 235/236: the statement "…, but they are not specific for brachyfacial patients" is misleading. Please specify (regarding the information on facial types / vertical values given or missing in the cited literature).
- line 237: while Gunay et al. treated late adolescent patients, your patient sample was around the pubertal growth spurt. Please discuss the impact of the maturation stage on Class II treatment more in detail.
- line 265: as you write in line 273, the follow-up period of one year is short term. Therefore, I do not agree with the statement that the Austro appliance is able to modify vertical dimensions in the long term, based on the available literature and present study results.
Author Response
REPLY TO REVIEWER # 3
Thank you very much for your comments and the review you made of our research.
- Concern of the reviewer:
“The title and even Material and Method´s section are suggestive of two separate treatment phases: first the Austro appliance, followed by Multibracket appliance. However, in the discussion section (line 208) it is stated that all patients of the treated group wore the the Austro appliance the whole treatment time. I guess this information should (1) be placed in the Material and Method´s section and (2) lead to a modification of the manuscript title.”
Our response:
Done. This information has been placed in the Material and methods section. We think that the current manuscript title is compatible with these modifications.
- Concern of the reviewer:
“Abstract:
line 21: the minus signs between the number of boys and girls should be removed.
line 30: I guess a "d" is missing at the end of "LAFH distance increase.”
Our response:
Done.
- Concern of the reviewer:
“Introduction:
line 41: I guess again it should be "considered" instead of "considerer".
Our response: Done.
line 52: There is a much larger amount of studies dealing with the effect of Herbst appliance treatment on the mandibular plane angle than the two studies mentioned.
Our response: We agree with the reviewer, nevertheless only the most specific studies describing vertical changes in brachyfacial Class II patients have been included in our study.
- Concern of the reviewer:
“Material and Methods:”
I appreciate that the severity of Class II molar relationship is given in contrast to many studies which are keeping this major information secret. Nevertheless, the inclusion criteria regarding occlusal amount of Class II is set very low (one-fourth cuspid width). Had all treated patients a symmetric amount of Class II molar relationship pretreatment?
Our response: only patients with symmetric Class II molar relationship were included. This information has been added in the Inclusion criteria section.
What was the mean (SD) amount of Class II molar relationship in the present sample? Please add this information.
Our response: Done. This information has been added in the revised version of the manuscript (“mean amount of Class II molar relationship was 6.5 ± 2.23 mm”).
As mentioned above, please add the information of treatment length (mean, SD) and the fact that the Austro and Multibracket appliance were worn parallel.
Our response: Done. This issue has been clarified in the revision version of the manuscript, as commented in response to reviewer 2.
Please add the information, if class II elastics were worn during the multibracket phase.
Our response: Done. This information has been added at the end of 2.4. Class II malocclusion treatment protocol section, in the material and method (“No class II elastics were worn during the multibracket phase”).
- Concern of the reviewer:
“Results:”
all tables: please make sure that all parameters are named in English language (interincisal angle).
Our response: Done, tables have been revised and all parameters are now named in English language
if T1 records are the pretreatment records, why did you show a picture with the Austro appliance in situ? And why was it removed for the fourth picture after eruption of the lower right canine (large overjet, canines about ½ cusp widths class II relationship)
Our response: Sorry, there were several errors in this section. We have renamed and rearranged the figures for greater understanding.
why did you show a picture with the Austro appliance in situ?
Our response: Figure 4 has been modified. Incorrect pictures (frontal and lateral intraoral) have been replaced by the correct ones.
“And why was it removed for the fourth picture after eruption of the lower right canine (large overjet, canines about ½ cusp widths class II relationship)”
Our response: Figure 3 has also been revised and modified correcting these errors.
- Concern of the reviewer:
“Discussion:”
line 208: see above
lines 223-224: as mentioned above, there are studies on the stability of treatment results of fixed functional appliances dealing with vertical dimensions.
Our response: We agree with the reviewer, nevertheless specific studies on the stability of treatment results of fixed functional appliances dealing with vertical dimension in brachyfacial subjects are scarce.
225-229: as the present study found/assessed no specific vertical changes of the maxilla, the discussion of the two study results is unnecessary.
Our response: The discussion of the two study results about vertical changes of the maxilla has been removed.
235/236: the statement "…, but they are not specific for brachyfacial patients" is misleading. Please specify (regarding the information on facial types / vertical values given or missing in the cited literature).
Our response: These sentence has been modified as follow “…nevertheless, comparisons with the results obtained in our study have to be made with caution, because these studies included patients with different facial growth pattern (i.e. meso and brachyfacial patterns)”.
line 237: while Gunay et al. treated late adolescent patients, your patient sample was around the pubertal growth spurt. Please discuss the impact of the maturation stage on Class II treatment more in detail.
Our response: A new paragraph has been added in the discussion section “Subjects of both groups were between the CS3 and CS4 stage of skeletal maturation, which is when the peak of growth takes place. Literature concerning treatment timing for Class II malocclusion with functional appliances has revealed that the treatment is most effective when it includes the peak in mandibular growth [15]”
line 265: as you write in line 273, the follow-up period of one year is short term. Therefore, I do not agree with the statement that the Austro appliance is able to modify vertical dimensions in the long term, based on the available literature and present study results.
Our response: This sentence has been modified as follow “…The greater correction achieved in skeletal Class II, as well as its ability to modify vertical dimensions at least one year after treatment and retention period, compared to other functional devices, lead us to consider the Austro Repositioner as an effective option for the treatment of skeletal Class II patients with a mandibular origin and brachyfacial growth pattern.”

Round 2
Reviewer 2 Report
The article IJERPH-135682 entitled” Stability of Class II malocclusion treatment with the
Austro Repositioner followed by fixed appliances in brachyfacial patients” has been greatly improved after major revision.
Specific
- As the authors mentioned that “the lower incisors make contact with the anterior area of the acrylic resin wedge and prevent the occlusion of the posterior teeth,” are there any training instructions in the protocol (how long to maintain this motion per day、 contraindication? …..etc.) of the Autro Repositioer needed to provide for teenagers?
- Inclusion /Exclusion criteria: Are patients with TMJ disorder belong to inclusion or exclusion criteria?
Author Response
Dear reviewer,
Thanks for your comments. We proceed to answer the following questions.
Specific
- As the authors mentioned that “the lower incisors make contact with the anterior area of the acrylic resin wedge and prevent the occlusion of the posterior teeth,” are there any training instructions in the protocol (how long to maintain this motion per day、 contraindication? …..etc.) of the Autro Repositioer needed to provide for teenagers?
Authors Response: The special Austro Repositioner design for brachyfacial pattern prevent the occlusion of the posterior teeth when patients close the mouth, so no training instructions are needed. However, the informed consent included detailed information on the mechanism of action of the device.
- Inclusion /Exclusion criteria: Are patients with TMJ disorder belong to inclusion or exclusion criteria?
Authors Response: Done, Patients with TMJ disorder belong to exclusion criteria. This information has been included in the Material and method section, 2.3. Exclusion criteria
